# SATE: A TWO-STAGE APPROACH FOR PERFORMANCE PREDICTION IN SUBPOPULATION SHIFT SCENARIOS

## ABSTRACT

Subpopulation shift refers to the difference in the distribution of subgroups between training and test datasets. When an underrepresented subgroup becomes predominant during testing, it can lead to significant performance degradation, making performance prediction prior to deployment particularly important. Existing performance prediction methods often fail to address this type of shift effectively due to their usage of unreliable model confidence and mis-specified distributional distances. In this paper, we propose a novel performance prediction method specifically designed to tackle subpopulation shifts, called Subpopulation-Aware Two-stage Estimator (SATE). Our approach first estimates the subgroup proportions in the test set by linearly expressing the test embedding with training subgroup embeddings. Then, it predicts the accuracy for each subgroup using the accuracy on augmented training set, aggregating them into an overall performance estimate. We provide theoretical proof of our method's unbiasedness and consistency, and demonstrate that it outperforms numerous baselines across various datasets, including vision, medical, and language tasks, offering a reliable tool for performance prediction in scenarios involving subpopulation shifts.

## 1 INTRODUCTION

In the training and deployment of machine learning models, it is common to encounter shifts in data distribution (Shen et al., 2021). Such distributional discrepancies often result in degraded performance, making performance prediction prior to deployment particularly essential, especially in high-stakes domains like finance and medicine where the cost of errors is substantial.

A performance prediction method, also known as unsupervised accuracy estimation, typically takes in labeled training data, trained model and unlabeled test data. Its goal is to produce a direct or indirect measure of accuracy on test data. This serves not only as a confidence estimate but also aids in discerning which models are more suitable for specific datasets or which datasets are more compatible with a given model (Yu et al., 2024). This matching capability is even more important with an ever-growing number of models and algorithms to date.

Previous researchers commonly evaluate their performance prediction methods using two types of distribution shifts: synthetic shifts, where test datasets are generated through artificial perturbations (Hendrycks & Dietterich, 2019), and natural shifts, such as training on ImageNet (Deng et al., 2009) and testing on ImageNet-v2 (Recht et al., 2019). However, these types of distribution shifts do not encompass all scenarios encountered in real-world applications.

One underexplored type of shift in the field of performance prediction is the subpopulation shift. It refers to the difference in the training and testing distributions in terms of how well-represented each subpopulation is (Sagawa et al., 2020; Santurkar et al., 2020; Yang et al., 2023). Typically, subgroups are divided by labels and attributes. Significant performance degradation may occur when a subgroup that is underrepresented during training becomes prevalent while testing, making performance prediction before deployment especially necessary. Also, as highlighted in Yang et al. (2023), no single method remains state-of-the-art across all types and degrees of subpopulation shifts. indicating that it is improper to trust certain model without considering the test data.

In our work, we propose a performance prediction method specifically designed to address subpopulation shift called Subpopulation-Aware Two-stage Estimator (SATE). It has demonstrated superior

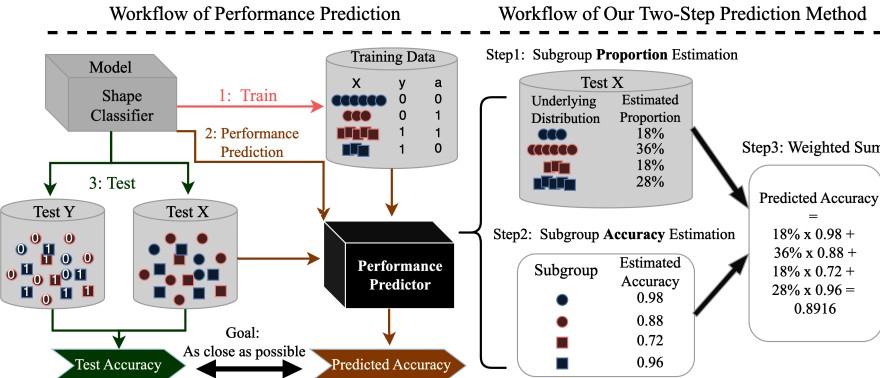

Figure 1: The workflow for predicting model performance under subpopulation shift conditions. Here the label is shape (circle vs square) and the attribute is color (red vs blue). Our method is decomposed into two main stages: subgroup proportion estimation and subgroup accuracy estimation. First, we estimate subgroup proportions by linearly express the test embedding. Second, subgroup accuracies are predicted using the augmented training data. The final predicted accuracy is the weighted average of these subgroup accuracies.

performance over multiple baselines on several classical subpopulation shift datasets including vision, medical, and language tasks. Our approach leverages attribute information and decomposes the performance prediction process into two steps. Firstly, we use the average embeddings of each training subgroup to linearly express the overall average embedding of the test data, thereby obtaining an estimation of the subgroup proportions. Secondly, we estimate the accuracy of each subgroup using the augmented training data. Finally, we obtain the overall predicted accuracy through a weighted average of these subgroup accuracies. Our main contributions are as follows:

1. We propose the first unsupervised performance prediction method specifically designed for subpopulation shift and demonstrate through experiments that it outperforms numerous baselines across multiple datasets. In settings with both subpopulation shift and covariate shift, our method improves the Pearson's correlation coefficient from below 0.74 to above 0.84, exceeding the best baseline.

2. We prove the unbiasedness and consistency of our method (under the presence of validation data or specific assumptions). Additionally, through experimental validation, we discovered a linear relationship between in-distribution accuracy and accuracy on an augmented training set. We use this insight to perform performance prediction without accessing additional data.

3. To the best of our knowledge, we are the first to address the problem of unsupervised performance prediction in NLP tasks. We highlight the challenges in designing synthetic datasets for NLP and demonstrate that a simple synthetic dataset design using large language models is effective for unsupervised performance prediction in NLP tasks.

## 2 PRIOR WORK

**Out-of-Distribution Performance Prediction.** Out-of-Distribution (OOD) Performance Prediction is an important research theme to characterize the OOD behavior of machine learning models. Its primary goal is to assess whether a machine learning model has good OOD generalization capabilities and to determine where it can perform well only with unlabeled test data (Yu et al., 2024). In this work, we focus on unsupervised performance prediction which means predicting the performance without relying on prior results from other datasets. This problem is also called unsupervised accuracy estimation (Diamantidis et al., 2000), since the most commonly used metric for a classifier's performance is accuracy.

Follow Yu et al. (2024), most previous performance prediction methods can be categorized into three types. (1) Model Output Property-based: Methods such as ATC (Garg et al., 2022), DoC (Guillory et al., 2021) and NI (Ng et al., 2022) predict performance based on the model's output (e.g. confidence) on the test data. (2) Distribution Discrepancy-based: Methods like Lu et al. (2023) , Yu et al. (2022) assess performance by evaluating some kinds of distance (e.g. Wasserstein Distance)

between the training and test data. (3) Model Agreement-based: Methods such as Baek et al. (2022) and Chen et al. (2021) predict performance by examining the output invariance of multiple slightly varied models (e.g. trained with different random seed). Note that NAC (Liu et al., 2024) is designed for the detection or evaluation of OOD models without test data and Deng & Zheng (2024) needs additional data to supervise the accuracy estimator, which are inconsistent with our setting.

Our algorithm differs from these approaches in several key ways. First, our approach does not rely on the model's output on the test data. Second, rather than computing the distance between overall distributions, our method focuses on linearly expressing the test set on the subgroup level. Finally, in contrast to model agreement-based methods, our method does not necessitate any extra model training. Recently, a method using VLM to extract priors for assisting failure detection has been tested on several subpopulation shift datasets (Subramanyam et al., 2024). Our approach differs by emphasizing dataset-level accuracy prediction, rather than estimating the probability of individual sample misclassification. Also, their method is limited to image datasets, whereas ours is not restricted.

**Subpopulation Shift.** In the context of Subpopulation Shift, each data point contains several attribute information $a$ in addition to input $x$ and label $y$. The entire dataset can be divided into multiple discrete subpopulations based on the combination of labels and attributes. However, the proportion of each subgroup may differ between the training and test datasets, causing one or more of Spurious Correlation, Attribute Imbalance, Class Imbalance, or Attribute Generalization (Yang et al., 2023). These types of subpopulation shifts will lead to performance degradation on the test dataset. Various methods have been studied to address this problem, including subgroup-based methods like GroupDRO (Sagawa et al., 2020) and IRM (Ahuja et al., 2020), data augmentation-based methods like Mixup (Zhang et al., 2018), reweighting-based methods like Megahed et al. (2021) and several two-stage methods such as JTT (Liu et al., 2021), CRT (Kang et al., 2019) and DFR (Izmailov et al., 2022). These approaches aim to improve model robustness and ensure consistent performance across different subgroups within the data. There have also been some methods capable of handling subpopulation shift without attribute annotations (Hong et al., 2024; Stromberg et al., 2024), but they all use some technique (optimal data partitioning, regularized annotation of domains) to perform their own subgroup partitioning, therefore they are still within our framework.

A model that can better handle subpopulation shift overall may exhibit a high Worst Group Accuracy (WGA) (Sagawa et al., 2020) because the test data could experience unpredictable changes and underrepresented groups may become major groups. Nevertheless, overall accuracy on the test dataset remains crucial. For instance, in a medical diagnosis application, a model must maintain a high overall accuracy to ensure reliable diagnostic results for the entire patient population. Overall accuracy is even more important in performance prediction's context because we already know where the model will be deployed on (the test data).

**Data Augmentation.** The goal of data augmentation is to enhance the diversity of the training set without collecting additional samples, thereby improving the model's generalization ability. Many easy-to-use and effective data augmentation methods are popular in computer vision (CV), such as cropping and flipping (Shorten & Khoshgoftaar, 2019). Thus, synthetic shifts in CV datasets are well-designed, and most performance prediction papers use self-designed (Deng et al., 2021) or existing synthetic datasets like ImageNet-P (Hendrycks & Dietterich, 2019) as their test sets.

However, as discussed in Feng et al. (2021), Shorten et al. (2021) and Pellicer et al. (2023), in the Natural Language Processing (NLP) field, due to the discrete nature of language and the difficulty in ensuring label invariance, data augmentation methods are relatively limited. Due to the above data restrictions, to the best of our knowledge, no unsupervised performance prediction method has yet been tested on language datasets. Note that Xia et al. (2020) and Srinivasan et al. (2021) require the performance results of a language model on several other datasets to run a regression for performance prediction, which is inconsistent with our setting where no prior historical information is available. Rychalska et al. (2019) and Talman et al. (2022) evaluated language models on corrupted datasets, but neither attempted to predict their performance in advance. Therefore, we believe that exploring performance prediction in the NLP domain is both novel and challenging.

## 3 PROBLEM SETUP

**Notations.** Following Yang et al. (2023) and Yu et al. (2024), we denote the input space as $\mathcal{X}$, output space as $\mathcal{Y}$ and the attribute space as $\mathcal{A}$. Considering the discrete case, $|\mathcal{Y}| = c, |\mathcal{A}| = m$. Then we define the subpopulations by a mapping $\mathcal{A} \times \mathcal{Y} \to \mathcal{G}$. $|\mathcal{G}| = c \cdot m$ is the number of subgroups. In our problem, we have a training set $S$ and a test set $T$, each sample can be represented by $\boldsymbol{z} = (\boldsymbol{x}, y, a)$, where $\boldsymbol{x}, y, a$ are random variables from $\mathcal{X}, \mathcal{Y}, \mathcal{A}$ respectively. The information available to us consists of $\boldsymbol{x}_S, y_S, a_S$ and $\boldsymbol{x}_T$. $S$ can be split into $c \cdot m$ subsets $(S_1, S_2, \cdots, S_{c \cdot m})$ by combinations of $y$ and $a$ . $T$ also consists of $c \cdot m$ subsets $(T_1, T_2, \cdots, T_{c \cdot m})$, however we cannot separate these subsets because the division of subpopulations on the test sets are unknown.

We define the model under evaluation as $f_\theta : \mathcal{X} \to \mathcal{Y}$, and we assume that it can be decomposed into two parts, the featurizer $f_{\theta_F} : \mathcal{X} \to \mathcal{H}$ and the classifier $f_{\theta_C} : \mathcal{H} \to \mathcal{Y}$, where $\mathcal{H} \in \mathbb{R}^d$ is the embedding space of dimension $d$.

The convergence in probability is denoted as $\xrightarrow{P}$. We call $\hat{\phi}$ a consistent estimator of $\phi$ if

$$\hat{\phi} \xrightarrow{P} \phi \Leftrightarrow \lim_{n \to \infty} \Pr\left(|\hat{\phi} - \phi| \geq \epsilon\right) = 0, \forall \epsilon > 0$$

We define the probability distribution as $\boldsymbol{x}_S \sim P_{src}$ and $\boldsymbol{x}_T \sim P_{tar}$. The train and test distribution are both mixtures of group-wise distributions. Let $g \in \mathcal{G}$ be a subgroup index and $\boldsymbol{x}_g$ be a random variable corresponding to a sample from subgroup $g$, such that $\boldsymbol{x}_g \sim P_g$. We have:

$$P_{src} = \sum_{g \in \mathcal{G}} \alpha_g P_g, P_{tar} = \sum_{g \in \mathcal{G}} \beta_g P_g \tag{1}$$

where $\alpha, \beta \in \mathbb{R}^{|\mathcal{G}|}, \sum \alpha = \sum \beta = 1$, they represent the subgroup proportions. i.e. For the same subgroup $i$, $S_i$ and $T_i$ follows the same distribution, the distribution shift in $S$ and $T$ is caused by different proportions of subgroups.

**Metrics.** Model $f_\theta$'s underlying accuracy on $T$ is denoted as $Acc_T \in \mathbb{R}$, and we define a loss function $l : \mathbb{R} \times \mathbb{R} \to \mathbb{R}$ describing the dissimilarity between the predicted output and the ground-truth accuracy. E.g. Mean Absolute Error (MAE). The performance prediction function is defined as $h(f_\theta, S, T) \to \mathbb{R}$. The goal of a direct accuracy prediction method is to find $h^*$ so that

$$h^* = \arg\min_h \mathop{\mathbb{E}}_{T \sim P_{tar}} [l(h(f_\theta, S, T), Acc_T)] \tag{2}$$

$Corr$ refers to a measure of relationship between two random variables, such as Pearson's Correlation Coefficient or Spearman's Rank Correlation Coefficient. The goal of a indirect performance prediction method is to find $h^*$ so that

$$h^* = \arg\max_h [Corr(h(f_\theta, S, T), Acc_T)] \tag{3}$$

In the calculation above, if we only need to know which datasets are more compatible with a single model, which previous works mainly focus on, then only $T$ provides randomness while $S$ and $f_\theta$ are fixed. If we need to compare multiple models together, which we take into consideration, then we should also take expectation over $S$ and $f_\theta$.

## 4 PROPOSED METHOD

### 4.1 MOTIVATIONS

Spurious correlation is a common issue in subpopulation shift datasets (Geirhos et al., 2020; Ye et al., 2024) that may cause confidence-based prediction methods to fail. It refers to non-causal relationship between an attribute $a$ and the label $y$ in the training set that does not hold in the test set. Models that rely heavily on such non-causal attributes may make incorrect predictions with high confidence when applied to the test set, thus results in unreliable confidence estimates.

Another complicating factor is that many algorithms designed to address subpopulation shifts often involve different usage of training samples during optimization, examples including Sagawa et al. (2020), Megahed et al. (2021), Izmailov et al. (2022). As a result, the distribution of the training dataset may differ from the distribution the model actually fits. Therefore, methods based on the distance between training and test datasets may fail on these algorithms.

Inspired by the work of He et al. (2024), who demonstrated that a weighted combination of source domains can effectively align the target dataset, our method, SATE, takes advantage of prior domain information. Instead of relying on model confidence or overall distribution distance, SATE linearly expresses the test set with training subgroups, evading the limitations above, thus providing a more accurate and reliable approach to handling subpopulation shifts.

Furthermore, most subpopulation shift experiments assume the availability of a validation set to guide model selection and evaluation (Yang et al., 2023) (Izmailov et al., 2022). Our proposed method also has the advantage of offering flexibility in using a validation set, and we show that its inclusion can improve the performance predictions.

## 4.2 Algorithm Workflow

We decompose the performance prediction process into two steps: subgroup proportion estimation and subgroup accuracy estimation. The final output is a weighted average of subgroup accuracies.

---

**Algorithm 1** Two Stage Performance Prediction

---

**Require:** Labeled training data $S$, Unlabeled test data $T$, Trained model $f_\theta$, Certain data augmentation method
1: Initialize $\boldsymbol{H}_S \in \mathbb{R}^{d \times (c \cdot m)}, \boldsymbol{a} \in \mathbb{R}^{c \cdot m}$ and $\boldsymbol{w} = \mathbf{1}_{c \cdot m}$
2: Categorize $S$ into $S_1 \cdots S_{c \cdot m}$ based on labels and attribute.
3: **for** each subgroup $S_i$ in $S$ **do**
4:     Compute average embedding $\bar{\boldsymbol{h}}_{s_i} \leftarrow \frac{1}{|S_i|} \sum_{\boldsymbol{x} \in S_i} f_{\theta_F}(\boldsymbol{x})$
5:     $S_i' \leftarrow \text{DataAugmentation}(S_i)$    $\{S_i' \leftarrow V_i \text{ if validation set } V \text{ is available}\}$
6:     $Acc_{S_i'} \leftarrow \frac{1}{|S_i'|} \sum_{(\mathbf{x}, y) \in S_i'} \mathbb{I}(f_\theta(\mathbf{x}) = y)$   {subgroup accuracy estimation}
7:     $\boldsymbol{H}_S[:, i] \leftarrow \bar{\boldsymbol{h}}_{s_i}, \boldsymbol{a}[i] \leftarrow Acc_{S_i'}$    $\{\boldsymbol{X}[:, i] \leftarrow \boldsymbol{x} \text{ means assigning } \boldsymbol{x} \text{ to the i-th column of } \boldsymbol{X}\}$
8: **end for**
9: Compute average embedding of test set: $\bar{\boldsymbol{h}}_T \leftarrow \frac{1}{|T|} \sum_{\boldsymbol{x} \in T} f_{\theta_F}(\boldsymbol{x})$
10: Solve the linear equation $\boldsymbol{H}_S \cdot \boldsymbol{w} = \bar{\boldsymbol{h}}_T$   {subgroup proportion estimation}
11: Calculate the estimated overall accuracy $\hat{Acc}_T = \boldsymbol{a} \cdot \boldsymbol{w}$
12: **return** $\hat{Acc}_T$

---

**Estimating Subgroup Proportion.** For embedding vectors, we denote $\boldsymbol{h}_T = f_{\theta_1}(\boldsymbol{x}_T)$, $\boldsymbol{h}_{T_g} = f_{\theta_F}(\boldsymbol{x}_{T_g})$ and their distributions as follows, $\boldsymbol{h}_T \sim P_{\text{T-emb}}$, $\boldsymbol{h}_{T_g} \sim P_{\text{g-emb}}$. Follow equation 1, overall test embedding distribution is also a mixture: $P_{\text{T-emb}} = \sum_{g \in \mathcal{G}} \beta_g P_{\text{g-emb}}$. So we can decompose the expectation of test embedding group-wisely,

$$E(\boldsymbol{h}_T) = [E(\boldsymbol{h}_{T_1}), E(\boldsymbol{h}_{T_2}), \cdots, E(\boldsymbol{h}_{T_{c \cdot m}})] \cdot [\beta_1, \beta_2, \cdots, \beta_{c \cdot m}]^T \tag{4}$$

From the sample perspective, we define the average embedding for subgroup $S_g$ as $\bar{\boldsymbol{h}}_{S_g} = \frac{1}{|S_g|} \sum_{\boldsymbol{x} \in S_g} f_{\theta_F}(\boldsymbol{x})$ and the average embedding for whole test set as $\bar{\boldsymbol{h}}_T = \frac{1}{|T|} \sum_{\boldsymbol{x} \in T} f_{\theta_F}(\boldsymbol{x})$. Note that $\bar{\boldsymbol{h}}_{S_g}$ and $\bar{\boldsymbol{h}}_T$ can be obtained by feeding the data into the trained featurizer and computing their average. Then we can get $\boldsymbol{w}$, an estimator of $\boldsymbol{\beta}$, by solving the following linear equation:

$$\bar{\boldsymbol{h}}_T = [\bar{\boldsymbol{h}}_{S_1}, \bar{\boldsymbol{h}}_{S_1}, \cdots, \bar{\boldsymbol{h}}_{S_{c \cdot m}}] \cdot [w_1, w_2, \cdots, w_{c \cdot m}]^T \tag{5}$$

It can be solve algebraically or by gradient descendant with MSE loss function.

**Assumption 1.** $\boldsymbol{x}_{S_g}$ and $\boldsymbol{x}_{T_g}$ follow the same distribution $P_g$, $\forall g \in \mathcal{G}$.

**Assumption 2.** *Matrix $\boldsymbol{H}_S$ (defined in Algorithm 1) is column full rank, i.e. mean embeddings of different subgroups are linearly independent.*

Assumption 1 is mensioned in Section 3 and it's a common setting in the field of subpopulation shift (Yang et al., 2023). Assumption 2 is reasonable because $|\boldsymbol{h}| \gg c \cdot m$, for example, the dimension of resnet-50 (He et al., 2016) embedding is 2048 and subgroup numbers for Waterbirds (Wah et al., 2011) is 4.

**Theorem 1.** *Estimated weight $\boldsymbol{w}$ is an unbiased and consistent estimator of subgroup proportion $\boldsymbol{\beta}$ under Assumption 1 and 2.*

See Appendix E for detailed proof.

**Estimating Subgroup Accuracy.** If we have access to validation set $V \sim P_{val}$, $P_{val} = \sum_{g \in \mathcal{G}} \gamma_g P_g$, regardless of its subgroup distribution $\gamma$, we can easily get an accuracy estimator $\hat{Acc}_{T_g} = Acc_{V_g}$, which is the accuracy on corresponding subgroup in validation set. This estimator is unbiased and consistent because corresponding subgroups follow the same distribution ($\boldsymbol{x}_{T_i}, \boldsymbol{x}_{V_i} \sim P_i$). Otherwise, without validation set, we first perform data augmentation on the training set. The augmented training set is denoted as $S'$, and get the estimator $\hat{Acc}_{T_g} = Acc_{S'_g}$. The purpose of data augmentation is to eliminate the inflated accuracy resulting from the model having previously seen the training samples.

The transformation should be label and attribute preserving. For image tasks (Waterbirds, CelebA, CheXpert), we use one or more of the following transformations from torchvision.transforms: RandomResizedCrop, RandomHorizontalFlip, RandomRotation, and ColorJitter. For the language task (MultiNLI), inspired by Whitehouse et al. (2023), we utilize large language models to rewrite sentences without altering their attributes or labels. Specifically, we use the ChatGPT-3.5-turbo and Llama-3.1-405B for rewriting. See Appendix D for the detailed prompt. Note that we differ from Anaby-Tavor et al. (2019), as they use simpler language models that require fine-tuning and generate sentence from scratch after being prompted with a label, while we do not require fine-tuning and our prompt can help ensure that the corresponding attribute remains unchanged.

Combining two components above, the predicted accuracy is:

$$\hat{Acc}_T = [\hat{Acc}_{T_1}, \hat{Acc}_{T_2}, \cdots, \hat{Acc}_{T_{c \cdot m}}] \cdot [w_1, w_2, \cdots, w_{c \cdot m}]^T$$

### 4.3 JUSTIFICATION OF USING DATA AUGMENTATION ON THE TRAINING SET

In this section, we will show why it is reasonable for us to use accuracy on augmented training set $Acc_{S'}$ as an estimator of $Acc_{T_i}$.

**Augmentation on the Line.** Miller et al. (2021) proposed the idea of Accuracy-on-the-Line. They found there exists a linear relationship between in-distribution (ID) accuracy and out-of-distribution (OOD) accuracy on certain datasets. Holding train and test distribution fixed, varying model, hyperparameters, training duration etc. all result in the same linear trend. Other interesting findings include those by Izmailov et al. (2022), which reveal a linear relationship between overall accuracy and WGA and Baek et al. (2022), which proposed the Agreement-on-the-Line, demonstrating that the agreement of the ID and OOD models exhibits a linear relationship.

We present a new finding regarding the linear relationship on the datasets we explored, namely "Augmentation-on-the-Line": ID accuracy has a linear relationship with the accuracy on augmented training data. Figure (2) shows that for the same task and augmenting method, regardless of varying model architectures, optimization algorithms, or subgroups it belongs to, the linear trend remains nearly the same. Furthermore, if the augmentation method is correctly chosen, this linear trend can be very close to $y = x$ (the red dash line). Here we simply use untuned RandAug (Cubuk et al., 2020) (image datasets) and LLM-rewriting (language dataset) as augmenting methods.

**Theoretical Correctness.** The above experimental finding can be expressed by this assumption,

**Assumption 3.** *Augmentation on the line:*

$$E(Acc_{S'_g}) = kAcc_{T_g} + b$$

$$Acc_{S'_g} \xrightarrow{P} kAcc_{T_g} + b$$

*Where $k$ and $b$ represents the slope and bias of the above linear relationship, they are both fixed among different subgroups and models.*

**Theorem 2.** *Under Assumptions 1, 2 and 3, predicted accuracy $\hat{Acc}_T$ has a linear relationship with underlying test accuracy $Acc_T$.*

$$E(\hat{Acc}_T) = kAcc_T + b$$

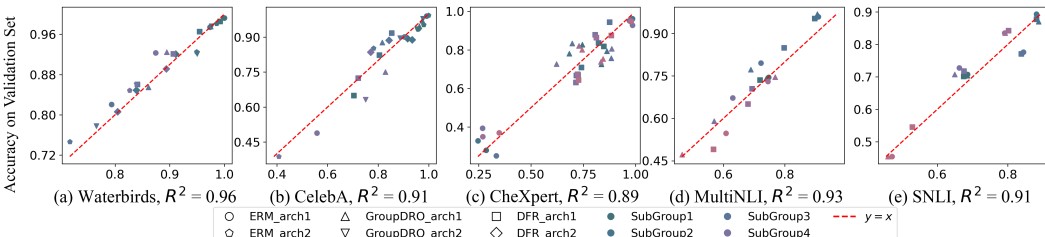

Figure 2: The linear correlation between ID accuracy (y-axis) and accuracy on augmented training data (x-axis). For image tasks, "arch1" represents ResNet50 and "arch2" is ViT; for language tasks, "arch1" stands for BERT. The red dashed line represents the ideal $y = x$ relationship. The linear trend remains consistent regardless of variations in model architecture or optimization algorithms, supporting the use of augmented training accuracy as a predictor for ID accuracy.

See Appendix E for detailed proof. This result is not affected by $\beta$ and $f_\theta$, allowing our approach to compare various models on multiple test sets together.

**Theorem 3.** *With the existence of validation data or the augmentation method is properly chosen so that k = 1, b = 0, predicted accuracy $\hat{Acc}_T$ is an unbiased and consistent estimator of underlying test accuracy $Acc_T$.*

## 5 EXPERIMENTS

### 5.1 TASKS AND MODELS

We conducted experiments on the following tasks: (1) Image Tasks: **Waterbirds** (Wah et al., 2011) and **CelebA** (Liu et al., 2015); (2) Medical Task: **CheXpert** (Irvin et al., 2019); (3) Language Tasks: **MultiNLI** (Williams et al., 2018) and **SNLI** (Bowman et al., 2015). See the Appendix for details.

For image tasks, we use two representative architectures, ResNet-50 (He et al., 2016) and Vision Transformer (ViT) (Dosovitskiy et al., 2020), supervised pretrained on ImageNet-1k (Deng et al., 2009). For language task, we use BERT-base-uncased (Devlin et al., 2019) as architecture. For each task-architecture combination, we use ERM (Vapnik, 1999), GroupDRO (Sagawa et al., 2020) and DFR (Izmailov et al., 2022) three algorithms to train the models.

### 5.2 BASELINES AND METRICS

**ATC-MC** and **ATC-NE** (Garg et al., 2022): First, we determine a confidence threshold based on overall training accuracy and then use this threshold to partition the test set. Test images with confidence above the threshold will be considered correct, otherwise incorrect. The threshold can also be based on negative entropy (NE). Its output is a direct estimator of test accuracy.

**DoC** and **DoE** (Guillory et al., 2021): Output the difference of confidence (or entropy) between training and test data. It's a indirect estimator of the accuracy gap between training and test.

**Neighborhood Invariance (NI)** (Ng et al., 2022): Deploy different data augmentation on test set and measure the invariance of model's output label. Here we use NI-RandAug, which uses augmentation from Cubuk et al. (2020). Its output is a indirect estimator of test accuracy.

**Datasets Design.** Unlike previous experiments on covariate shifts that can easily create a diverse range of test sets using corruption and perturbation, the design of test sets with subpopulation shifts is more challenging. The train-test split only provides a single test set for evaluation. This limits our ability to comprehensively compare model performance across different degrees and types of subpopulation shifts. Thus, to construct the test data, we designed 20 different subgroup distributions to simulate a wide range of diverse subpopulation shifts. Note that we only control the number of samples extracted from each subgroup, the extraction within a subgroup is still random. To construct the training data, we randomly sampled from data outside the test sets, with the distribution as similar as possible to the distribution of the original overall dataset. Details are provided in the appendix A.

**Metrics.** A good performance prediction method should give a test accuracy estimation $\phi_T$ highly correlated to the groundtruth test accuracy $Acc_T$. The following metrics are used for comparison, (1) **coefficient of determination ($R^2$)**: the goodness of linearly fitting $Acc_T$ with $\phi_T$. (2) **Mean Absolute Error (MAE)**: the error between $Acc_T$ and $\phi_T$ if used as direct estimator, or the error between $Acc_T$ and fitted value if used as indirect estimator. (3) **pearson's correlation coefficient**.

## 5.3 RESULTS

**Test Sets Characteristics.** We first propose a quantitative metric for subpopulation shift and demonstrate that the test sets we designed effectively simulate different degrees and types of subpopulation shifts. Previous work utilizes entropy and mutual information to quantify the degree of different subpopulation shifts (Yang et al., 2023), focusing solely on the imbalance within a single dataset. However, when considering model performance degradation, a metric that measures the difference between two datasets is more useful. To address this, we propose a new set of quantification metrics using Jensen-Shannon (J-S) divergence. We employ the divergence in $P(y), P(a), P(y|a)$ as metrics for Class Imbalance, Attribute Imbalance, and Spurious Correlation, respectively. To be specific, $P(y)$ of training set and $P(y)$ of test set are two discrete probability distributions, their J-S divergence is the metric of class imbalance. See Appendix C for detailed calculation procedure.

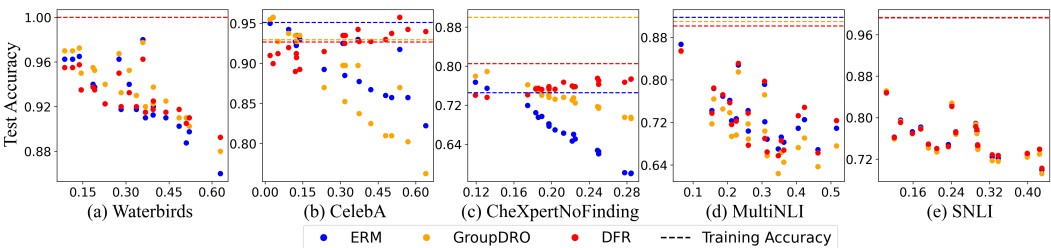

Figure 3: The relationship between the Jensen-Shannon (J-S) divergence of $P(y|a)$ (x-axis) and the test accuracy (y-axis) under different subpopulation shifts. Each point represents a unique test set, while the training data remains constant. A strong negative correlation is observed, particularly under ERM algorithm, where higher divergence often leads to lower test accuracy. This emphasizes the relevance of J-S divergence as a metric for subpopulation shifts and validates the efficacy of the manually designed test sets in simulating a diverse range of subpopulation shifts for testing.

Figure (3) clearly demonstrate that divergence in $P(y|a)$ exhibits a strong negative relationship with ERM test accuracy across all the tasks. However, specially designed algorithms, such as GroupDRO and DFR, may mitigate this accuracy degradation. Results are similar for $P(y)$, but divergence in $P(a)$ do not lead to degraded performance (in appendix B). i.e. spurious correlation and class imbalance will cause performance degradation in these settings, which matches with our intuition. This result not only shows that J-S divergence is a reasonable metric for subpopulation shift but also illustrates that our manually designed test sets effectively simulate a diverse range of distribution shifts and the degree of shifts are relatively even.

**Comparisons as Indirect Estimator.** An indirect performance prediction metric (defined in Equation 3) has two key capabilities. (1) **Model Comparison**: determining which model is best suited for a specific test set, i.e. in Equation 3, view $T$ as a constant and $f_\theta, S$ as random variables. (2) **Test Set Comparison**: identifying which test set is most suitable for a particular model, view $S, f_\theta$ as constants and $T$ as a random variable. Therefore, our results are presented separately for these two capabilities in Figure 5a. Note that our method have two versions of output: if augmented training data is used ($\hat{Acc}_{T_g} = Acc_{S'_g}$), the output will be denoted as SATE; if validation data is used ($\hat{Acc}_{T_g} = Acc_{V'}$), it will be denoted as SATE-val.

For model comparison, we perform regression on 6 models 20 times and take the average. For test set comparison, we perform regression on 20 test sets 6 times and take the average. "Regression" here means to regress the underlying accuracy on each method's output, for DoC and DoE, we regress $\Delta Acc$ on them, then add back training accuracy to get their predicted accuracies. Note that Neighborhood Invariance (NI) cannot be applied to NLP datasets, thus its bar is omitted.

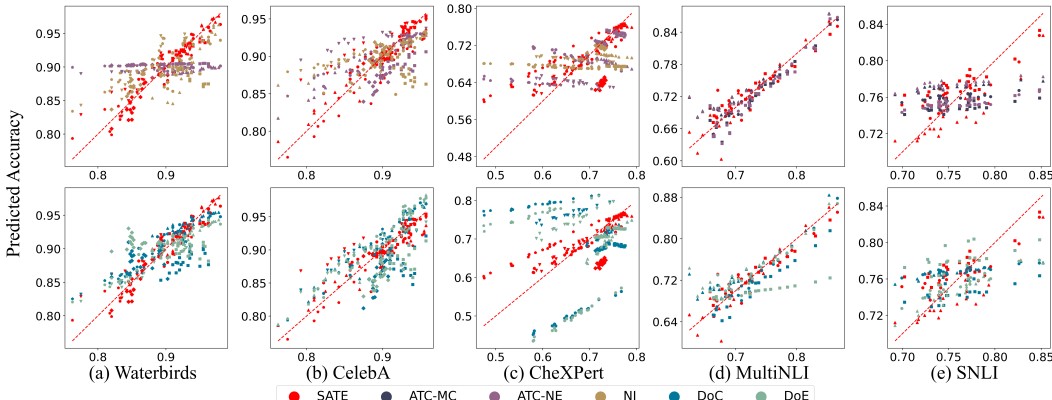

Figure 4: The relationship between predicted accuracy (y-axis) and actual accuracy (x-axis) on test set. For clarity, we separately present ATC-MC/NE and NI baselines in the top row and DoC/DoE in the bottom row. SATE results are presented in both rows. The color of each point represents the predictor used, while the shape indicates the model structure and training algorithm (consistent with Figure 2). The red dashed line represents $y = x$; the closer the distribution aligns with this line, the better the predictor. It is clear that SATE provides estimates with the lowest bias and variance across all settings.

**Comparison as a Direct Estimator.** Since ATC can be used as direct metrics, we also compare the results by directly calculating the MAE without regression. Note that here we consider $6 \times 20$ results together to demonstrate the comprehensive capability of performance predictors. Results are shown in Figure 5b. Our method consistently achieves lower MAE across all tasks and shows significant improvements on medical and language tasks.

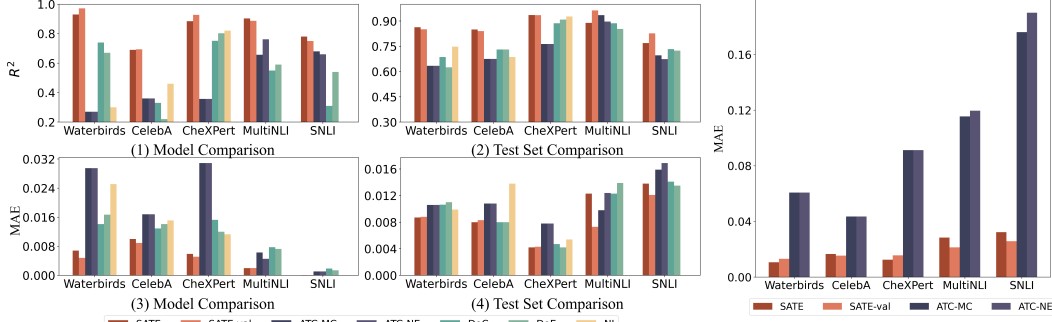

(a) Results of two distinct capabilities. Subplots (1)(3) are model comparison while (2)(4) are test set comparison. (1)(2) are measured by $R^2$ (higher is better) and (3)(4) are measured by MAE (lower is better). Our method outperforms baselines in both tasks, especially in model comparison. While each baseline fails significantly on at least one dataset.

(b) Comparison of MAE without regression. Our method consistently outperforms ATC, achieving significantly lower error. The inclusion of validation set further enhances our method's performance in most tasks.

**Real-World Shift** We aim to simulate test sets that more closely reflect real-world distribution shifts by introducing both subpopulation shift and covariate shift. Therefore, we added five types of perturbations (Fog, Blur, Noise, Contrast and Brightness) to the original 20 test sets. For each type of perturbation, two degrees are tested, forming 60 test sets (including the original 20). The three algorithms are still applied, and in the end, we regress 180 results to evaluate the overall capability of each performance prediction algorithm, which is shown in Table 1.

# 6 CONCLUSION

Our paper proposes a novel algorithm for model performance prediction under subpopulation shifts. We break down the performance prediction into two steps: proportion estimation and accuracy esti-

Table 1: Simulation of real-world distribution shifts on Waterbirds dataset, regressed over 3 algorithms and 60 test sets to evaluated predictors' overall capability. The additional 40 test sets were created by introducing two degrees of each corruption on the original 20 test sets. SATE uniformly outperforms all baselines under the presence of both subpopulation shifts and covariate shifts.

| Corruptions | Metrics | ATC-MC | DoE | NI | SATE |
|---|---|---|---|---|---|
| Fog | Correlation Coefficient ↑ | 0.330 | 0.447 | 0.644 | **0.841** |
| | $R^2$ ↑ | 0.109 | 0.199 | 0.414 | **0.707** |
| | MAE ↓ | 0.0174 | 0.0165 | 0.0133 | **0.0093** |
| Gaussian Blur | Correlation Coefficient ↑ | 0.596 | 0.557 | 0.586 | **0.876** |
| | $R^2$ ↑ | 0.355 | 0.311 | 0.343 | **0.767** |
| | MAE ↓ | 0.0219 | 0.0223 | 0.0207 | **0.0125** |
| Gaussian Noise | Correlation Coefficient ↑ | 0.389 | 0.703 | 0.669 | **0.908** |
| | $R^2$ ↑ | 0.152 | 0.494 | 0.448 | **0.824** |
| | MAE ↓ | 0.0202 | 0.0141 | 0.0151 | **0.0089** |
| Contrast | Correlation Coefficient ↑ | 0.428 | 0.736 | 0.616 | **0.910** |
| | $R^2$ ↑ | 0.183 | 0.543 | 0.379 | **0.827** |
| | MAE ↓ | 0.0332 | 0.0147 | 0.0163 | **0.0090** |
| Brightness | Correlation Coefficient ↑ | 0.399 | 0.727 | 0.668 | **0.899** |
| | $R^2$ ↑ | 0.160 | 0.528 | 0.447 | **0.808** |
| | MAE ↓ | 0.0199 | 0.0137 | 0.0146 | **0.0086** |

mation, effectively leveraging subgroup domain information to enhance our predictions. Extensive experiments over multiple datasets have demonstrated that our model outperforms the baselines in overall performance and it exhibits noticeably smaller bias when used as a direct metric, especially when used in model comparison. In scenarios with both covariate shift and subpopulation shift, which are closer to real-world conditions, our method also consistently outperforms all baselines. Additionally, it is evident that the addition of a validation set also leads to a slightly better performance in practice. This gives our method a greater advantage when validation data is available, as many other methods cannot directly utilize it.

## 7 LIMITATIONS

**Dependency on Attribute Annotations.** Our method relies on the availability of attribute annotations in the training set for subgroup division. In cases where such annotations are unavailable, our approach must be combined with unsupervised subgroup partitioning algorithms or require manual selection of a feature from $X$ as the attribute based on human knowledge. Furthermore, our method assumes that these attributes are accurate and complete. However, in real-world scenarios, attribute annotations may be noisy, incomplete, or biased, which could result in errors in both subgroup proportion estimation and performance prediction.

**Limited to Subpopulation Shifts.** Our approach is specifically designed to handle subpopulation shifts, operating under the assumption that the primary distributional changes stem from differences in subgroup proportions between the training and testing data. It cannot effectively quantify or address covariate shifts (changes in the overall feature distribution). Moreover, when strong covariate shifts occur together with subpopulation shifts, the test set embedding may no longer be able to represented as a linear combination of training subgroup embeddings, leading to inaccurate proportion estimates. As a potential direction for future work, integrating our method with other performance prediction techniques could create a more robust predictor capable of managing both subpopulation and covariate shifts effectively.

**Challenges with Unseen Subgroups.** The presence of unseen subgroups in the test set-subgroups that do not exist in the training set-can further increase estimation errors. To address this issue, we propose a lightweight method in appendix F for detecting the presence of unseen subgroups. This enhancement improves the adaptability of our method to domain generalization settings.

**Reproducibility Statement.** To ensure the reproducibility of our results, we used untuned and consistent random seeds during both model training and performance prediction. Our code is based on the implementation from SubpopBench (Yang et al., 2023), with the majority of model training parameters kept at their default settings. Additionally, the sample sizes drawn from each subgroup

for both the training and testing datasets are clearly documented (in Appendix A), and we also employed an untuned constant as random seed during the sampling process. Detailed code can be found in the supplementary material.

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

# A DATASET DETAILS

**Test Set Design.** Figures 6 and 7 show the distribution of 20 test sets we artificially designed. Each test set is represented by $c \times m$ grid, and the number of samples for each subgroup is marked in the square center, where darker color indicate a larger number of samples.

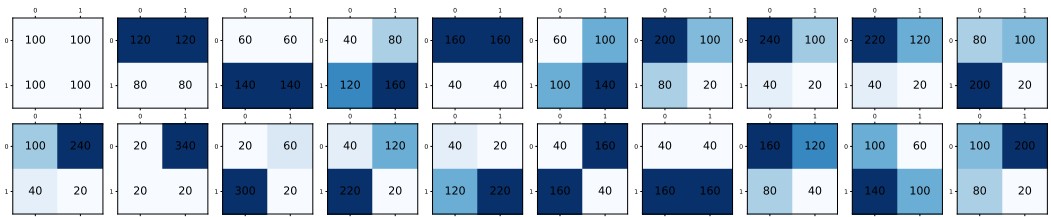

Figure 6: Subgroup distribution of test sets, for datasets with 4 subgroups (Waterbirds and CelebA).

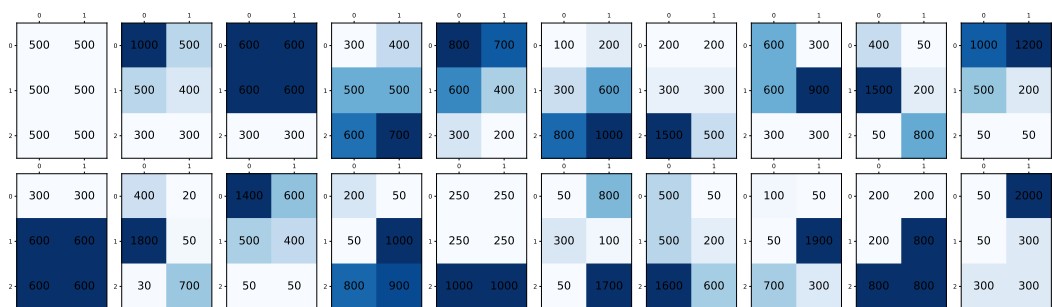

Figure 7: Subgroup distribution of test sets, for datasets with 6 subgroups (MultiNLI and SNLI).

Table 2 and 3 show some basic information about the datasets we used. Note that for SNLI, we consider a sentence has negation if one or more of the following words appears: no, never, nobody, nothing, not, none, nowhere, neither, nor. For other datasets, the annotations for $y$ and $a$ are consistent with those in Yang et al. (2023)

Table 2: Overview of the tasks we used in experiments.

|  | $|\mathcal{Y}|$ | $|\mathcal{A}|$ | meaning of $y$ | meaning of $a$ |
|---|---|---|---|---|
| Waterbirds | 2 | 2 | 1 if water-bird | 1 if water-background |
| CelebA | 2 | 2 | 1 if blond hair | 1 if male |
| CheXpert | 2 | 6 | 1 if no anomalies found | different ethnic groups |
| MultiNLI | 3 | 2 | neutral, contradiction, or entailment | 1 if negation appears |
| SNLI | 3 | 2 | neutral, contradiction, or entailment | 1 if negation appears |

Table 4 specifies the degree of corruptions we used in real-world shift experiments (Table 1). 20 original test sets, 20 sets with corruption 1 and 20 sets with corruption 2 together form 60 test sets for evaluation of performance predictors.

Table 3: Group-wise number of samples.

|  | total | train |
|---|---|---|
| Waterbirds | 6220, 2905, 831, 1832 | 3000, 1400, 400, 900 |
| CelebA | 89931, 82685, 28234, 1749 | 8000, 7500, 3000, 500 |
| CheXpert | 68899,44917,5399,5173,44851,31229, 7170,4638,727,671,5170,3948 | 6889,4491,539,517,4485,3122, 717,463,72,67,517,394 |
| MultiNLI | 114909,22447,134821,3020,133215,3937 | 11373,2242,13228,316,11411,370 |
| SNLI | 181232,8470,188030,1188,189916,1316 | 3150,328,3219,77,3460,87 |

Table 4: Detailed degrees of corruptions used in real-world shift experiments (Table 1).

|  | Fog | Gaussian Blur | Gaussian Noise | Contrast | Brightness |
|---|---|---|---|---|---|
| Corruption1 | 0.1 | radius=1 | sigma=1 | 1.2 | 1.2 |
| Corruption2 | 0.2 | radius=2 | sigma=2 | 1.4 | 1.4 |

## B  ADDITIONAL RESULTS

**Test Sets characteristics.** Figure 8 and 9 demonstrate the relationship between J-S divergence of $P(y)$, $P(a)$ and test accuracy. In most settings there is a clear negative relationship between J-S divergence of $P(y)$ and the test accuracy, while $P(a)$ seems to have no relationship with it.

**Predicted subgroup proportions versus actual proportions.** We measured the dissimilarity between predicted subgroup proportions and actual proportions using Wasserstein distance and cross entropy. The results are shown in Table 5

Table 5: Dissimilarity between predicted subgroup proportions and actual proportions measured by Wasserstein Distance (WD) and Cross Entropy (CE).

|  | Waterbirds | CelebA | CheXpert | MultiNLI | SNLI |
|---|---|---|---|---|---|
| WD | $0.053 \pm 0.039$ | $0.039 \pm 0.031$ | $0.028 \pm 0.008$ | $0.049 \pm 0.019$ | $0.065 \pm 0.023$ |
| CE | $1.22 \pm 0.15$ | $1.19 \pm 0.16$ | $2.47 \pm 0.02$ | $1.66 \pm 0.25$ | $2.26 \pm 0.36$ |

**Compare as direct estimator.** Figure 10 shows the detailed comparison results of SATE and ATC when used as an direct estimator. The predictor's output is directly plotted on the x-axis without any regression. SATE has lower bias and variance in most settings, outperforming ATC.

## C  J-S DIVERGENCE

We use J-S Divergence as a quantitative metric for subpopulation shift. In this section, we will detail the calculation.

For example, consider the training distribution [100, 100, 100, 100] and the test distribution [200, 100, 50, 50]. Each number represents the number of samples in a subgroup.

**J-S Divergence of $P(y)$.** **(1) Combine:** Merge by $y$, the two distributions become [200, 200] and [300, 100]. **(2) Normalize:** After normalization, they become [0.5, 0.5] and [0.75, 0.25]. At this point, both distributions are in the form of discrete probability distributions. **(3) Compute:** Use the standard J-S divergence calculation method to compute the divergence between these two vectors, which yields the final result, which is 0.221 in this example. Note that J-S Divergence of $P(a)$ is similar to this one, only differs in that we should merge by $a$.

**J-S Diverdence of $P(y|a)$.** **(1) Group by $a$:** Calculate J-S Divergence of $P(y)$ for each $a \in \mathcal{A}$ respectively, denoted as $JSy_a$ **(2) Weighted sum:** J-S Divergence of $P(y|a) = \sum_{a \in \mathcal{A}} P_T(a) \cdot JSy_a$, where $P_T(a)$ means the proportion of samples with attribute $a$ within test set.

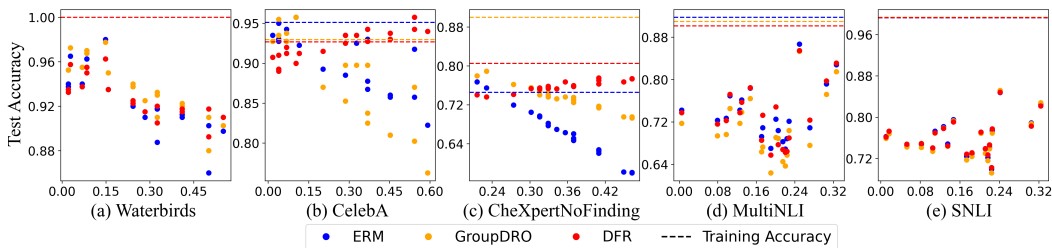

Figure 8: The relationship between the Jensen-Shannon (J-S) divergence of $P(y)$ (x-axis) and the accuracy on test datasets (y-axis) under different degree of subpopulation shifts. There exist a clear negative relationship between divergence in $P(y)$ and the test accuracy.

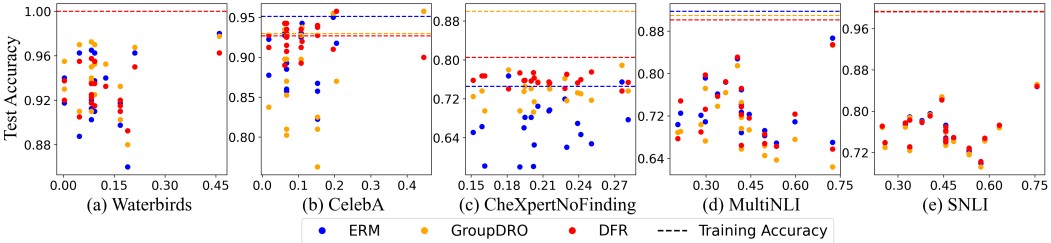

Figure 9: The relationship between the Jensen-Shannon (J-S) divergence of $P(a)$ (x-axis) and the accuracy on test datasets (y-axis) under different degree of subpopulation shifts. Their relationship is not significant.

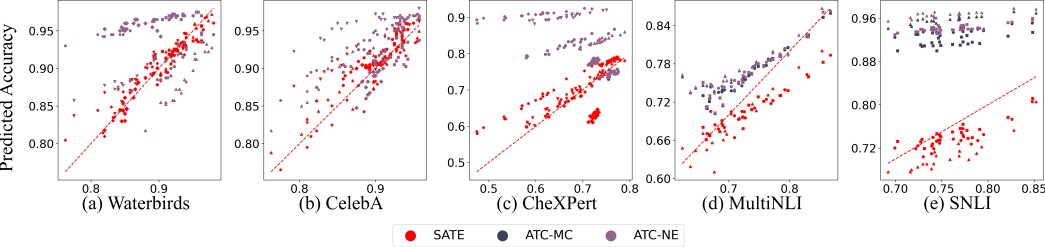

Figure 10: The relationship between predicted accuracy and actual test accuracy. Here we plot the predictor's output directly without doing any regression. Our method has lower bias and variance than ATC in most settings, indicating SATE is a better direct performance estimator in these cases.

## D LLM-AUGMENTING

**Prompt Part 1.** In MultiNLI and SNLI tasks, the relationship between two sentences is label $y$ and the exsistence of negation is attribute $a$, so the following prompt is designed to ensure that after rewriting, $y$ and $a$ still remain the same. We use ChatGPT-3.5-turbo for MultiNLI task and Llama-3.1-405b for SNLI task.

User:

```
Respectively rewrite the following two sentences, changing
their expression without altering the original meaning.
Also, ensure that the relationship between the meanings of
the two sentences remains unchanged (neutral, entailment
or contradiction).  Furthermore, make sure that if there is
negation in the original, it cannot be removed; if there is
no negation, it cannot be added.  Avoid using obscure words.
Each element should be a token.
```

**Prompt Part2.** This section shows the in-context learning we used during the LLM-augmenting.

User:

```
Fun for adults and children.  [SEP] Fun for only children.
```

Assistant:

```
Adults and children all consider it funny.  [SEP] Only
children consider it funny.
```

User:

```
You and I both fought him and he nearly took us.  [SEP]
Neither you nor myself have ever fought him.
```

Assistant:

```
You and I both faced him in battle and he nearly defeated
us.  [SEP] Neither you nor I have ever faced him in battle.
```

After that, the sentences to be rewrited will be provided to the LLM in the same format.

## E PROOFS

**Proof of Theorem 1.**

*Proof.* Assumption 1 guarantees

$$\boldsymbol{h}_{S_g}, \boldsymbol{h}_{T_g} \sim P_{\text{g-emb}}, \forall g \in \mathcal{G}$$

together with law of large numbers, we have

$$\bar{\boldsymbol{h}}_{S_g} \xrightarrow{P} E(\boldsymbol{h}_{S_g}) = E(\boldsymbol{h}_{T_g}), \forall g \in \mathcal{G}$$

$$\bar{\boldsymbol{h}}_T \xrightarrow{P} E(\boldsymbol{h}_T)$$

Assumption 2 ensures that Equation 5 has a unique solution. Take probability limit over both sides of Equation 5 and compare it Equation 4, we can get

$$\boldsymbol{w} \xrightarrow{P} \boldsymbol{\beta}$$

The proof for unbiasedness follows similar steps as the proof for consistency.

**Proof of Theorem 2.**

*Proof.* because

$$A\hat{c}c_T = \sum_{g \in \mathcal{G}} w_g A\hat{c}c_{T_g}, A\hat{c}c_{T_g} = Acc_{S'_g}$$

with Therorem 1 and Assumption 3,

$$E(A\hat{c}c_T) = \sum_{g \in \mathcal{G}} \beta_g E(Acc_{S'_g}) = k \sum_{g \in \mathcal{G}} \beta_g Acc_{T_g} + b = k Acc_T + b$$

## F  UNSEEN SUBGROUP DETECTION

Here we develop a lightweight method to detect unseen subgroups after the first step of SATE. We use Mean Square Error (MSE) of the linear decomposition as the indicator of the existence of unseen subgroups. Larger MSE indicates higher probability that the test set contains unseen subgroup.

We conducted experiments on the NICO++, a commonly used domain generalization benchmark, to evaluate our detection method. The experimental setup and findings are as follows:

**Benchmark Setup:**  We utilized the NICO++ (Zhang et al., 2022) dataset, focusing on $y \in \{0, 1, 2, 3, 4, 5\}$ and $a \in \{0, 1, 2, 3, 4, 5\}$, resulting in 36 subgroups in total. The training data followed the original split, where subgroup $(5, 4)$ was absent. While all 36 subgroups were present in the original test split.

**Test Sets:**  To simulate various conditions, we created 50 test sets, each comprising $k$ randomly selected subgroups from the original test set.

**Evaluation:**  We evaluate the effectiveness of detection by the Area Under the Curve (AUC) between the existence of unseen subgroup and the MSE of linear decomposition.

Table 6: the Area Under the Curve (AUC) between the existence of unseen subgroup and the MSE of linear decomposition.

| $k$ | 5 | 10 | 20 |
|-----|-----|-----|-----|
| AUC | 0.950 | 0.895 | 0.869 |

These results demonstrate that while using linear decomposition to estimate subgroup proportions, MSE is a reliable metric for detecting unseen domains. It consistently performs well when the number of subgroups in the test set becomes large ($k = 10, 20$), further extending the applicability of our method to domain generalization scenarios.

