# OpenReview forum: "SATE: A Two-Stage Approach for Performance Prediction in Subpopulation Shift Scenarios"
_ICLR.cc/2025/Conference — Submitted to ICLR 2025_

### Official Review · Reviewer_Vhym · 2024-10-16

**Soundness:** 3
**Presentation:** 1
**Contribution:** 2
**Rating:** 3
**Confidence:** 4

**Summary:**

The paper addresses how to predict the performance of an unlabeled test set in the presence of subpopulation shifts between the training and test sets. The authors propose a two-stage method. First, they estimate the proportions of different subpopulations in the test set by leveraging the average feature representation of all test samples and comparing it with the prototype features of each subpopulation in the training set. Next, they evaluate the performance of each subpopulation individually using a data-augmented version of the training set. Finally, the predicted overall test set performance is obtained by computing the weighted average of the subpopulation performances. The authors validate this approach with experiments on image and NLP datasets.

**Strengths:**

1. The study of performance prediction methods robust to distribution shifts is practical and meaningful.
2. The method proposed by the paper is straightforward and reasonable.
3. The authors provide the source code, which is highly commendable.

**Weaknesses:**

1. The writing of the paper should be improved, as the flow of logic is unclear in several parts. For example, the logic between the first four paragraphs of the introduction is confusing, and the same lack of clarity is present in the four paragraphs of section 4.1.
2. If I understand correctly, the terms subpopulation, subgroup, group, and subset in the paper are used interchangeably to convey the same meaning. This inconsistent terminology further increases confusion for the readers.
3. The theoretical part of the paper is trivial, lacking valuable insights in both the proof process and the results presented. I suggest that this part should not occupy such a significant portion of the manuscript and could potentially be removed from the main text altogether.
4. I have some concerns about the effectiveness of using a data-augmented training set. Modern image classification models typically employ a wide range of data augmentation techniques to enhance model performance. Therefore, the model should also perform well on augmented training images, especially given the simple geometric transformations like Crop, Flip, and RandomRotation used in the paper. I briefly reviewed the source code provided by the authors, and if I understand correctly, these augmentation techniques do not seem to be incorporated into the training process. This implies an assumption that appears to be rather unrealistic.
5. The baseline methods mentioned in Section 2, such as Distribution Discrepancy-based and Model Agreement-based approaches, do not appear to be compared in the experiments.
6. The authors emphasize spurious correlation in the motivation section, which raises a question for me: is the method aimed at addressing all types of subpopulation shifts, or is it specifically targeting spurious correlations? Based on my understanding, the former is correct. Therefore, what is the purpose of highlighting spurious correlation in this context?

Based on my current assessment, this paper is not sufficient for publication at ICLR. I will adjust my score accordingly based on the authors’ clarifications and modifications during the rebuttal phase.

**Questions:**

My questions that need clarification are included in the weaknesses section.

---

> ### Author Response · Authors · 2024-11-25
> **Response to Reviewer Vhym**
>
> Dear Reviewer Vhym,
>
> We thank you for your valuable feedback and constructive suggestions. Below, we address each of your comments in detail:
>
> > **W1.1**: The first four paragraphs of the introduction are confusing.
>
> **R1.1**: We would like to clarify the logic of the first four paragraphs of the introduction:
>
> - Paragraph 1: Highlights the importance of performance prediction.
> - Paragraph 2: Defines what performance prediction means.
> - Paragraph 3: Discusses the focus of previous performance prediction methods.
> - Paragraph 4: Identifies subpopulation shift as a research gap in this field.
>
> We have made some edits to the paper to make the structure more clear.
>
> > **W1.2**: Section 4.1 is confusing.
>
> **R1.2**: We acknowledge that Section 4.1 includes various content, and we could have used clearer logical connections to make the flow more explicit. We have revised the paper to improve clarity and better guide readers through the section. Briefly, the updated structure is as follows:
>
> - The first two paragraphs explain why current methods (confidence-based and distance-based) fail in subpopulation shift scenarios.
> - The third paragraph introduces the origin of our linear decomposition idea.
> - The final paragraph highlights the benefits of a method that can flexibly incorporate the validation set.
>
> Together, these elements establish the motivation for inventing SATE.
>
> > **W2**: Inconsistent terminology.
>
> **R2**: While some terms such as "subset," "subpopulation," "subgroup," and "group" convey similar meanings, we believe their usage does not cause confusion. To clarify:
> - The term “subset” is used **exclusively in the notation section** for mathematical clarity.
> - The term “subpopulation” is **always paired with the word "shift"** to align with established terminology in the field (e.g., "subpopulation shift").
> - We have **updated the paper and replace the word "group" with "subgroup"** to make terminology more clear.
>
> > **W3**: Theoretical part should be removed from the main text.
>
> **R3**: We agree with this suggestion. In the revised version, we will move the detailed proof to the appendix and only keep the assumptions and propositions in the main text.
>
> > **W4**: Data augmentation is not incorporated into the training process.
>
> **R4**: You are correct that data augmentation is not incorporated into the training process. Our implementation is built on SubpopBench, a widely used benchmark in the field, which does not include data augmentation in its codebase [1]. Similarly, ATC, one of the baselines in our experiments, also excludes data augmentation for datasets such as ImageNet, ImageNet-200, and the language tasks in their source code [2]. To ensure a fair comparison with prior work, we follow the same settings by excluding data augmentation during training.
>
> Additionally, incorporating data augmentation during training would not necessarily harm the performance of our method. Specifically:
>
> 1. When validation data is available, we do not rely on the accuracy of the augmented training set, as our approach leverages validation performance.
>
> 2. When validation data is unavailable, we could use a data augmentation method different from the one applied during training to conduct our approach.
>
> Due to time constraints, we will include additional experimental results on this topic in the final version of the paper.
>
> > **W5**: Some baselines are not compared in the experiments.
>
> **R5**: We agree that including additional baselines could strengthen the evaluation. However, we did not test distribution discrepancy-based and model agreement-based methods for the following reasons:
>
> - **Distribution Discrepancy-based Methods:** These methods rely on hidden features… are unsuitable for the Model Comparison task because they cannot distinguish between models that share the same featurizer (e.g. ERM and DFR). Moreover, algorithms such as GroupDRO, which optimize for the worst-group performance, make the model fit into a distribution that differs from the original training set distribution, making distributional distance measures misleading.
> - **Model Agreement-based Methods:** These methods require retraining the model multiple times, which introduces significant computational cost. Also, they assume full access to the model's architecture and training details, making them impractical for many real-world scenarios.
>
> > **W6**: Why mention spurious correlation?
>
> **R6**: Our method is designed to address all types of subpopulation shifts. Spurious correlation is mentioned specifically because it provides an intuitive example of how subpopulation shifts can cause confidence-based performance prediction methods to fail. This serves as a rationale for not relying on confidence as a predictor in our approach.
>
> [1] https://github.com/YyzHarry/SubpopBench
>
> [2] https://github.com/saurabhgarg1996/ATC_code

---

> > ### Comment · Reviewer_Vhym · 2024-11-29
> >
> > Thank you for your response. I will keep my score.

---

> > > ### Author Response · Authors · 2024-12-02
> > >
> > > Thank you for taking the time to review our response. We are happy to discuss any concerns that have not yet been addressed.

---

### Official Review · Reviewer_qi9w · 2024-11-04

**Soundness:** 2
**Presentation:** 3
**Contribution:** 2
**Rating:** 3
**Confidence:** 4

**Summary:**

This paper proposes SATE, a method for predicting test performance under subpopulation shift scenarios. The approach assumes access to test data but not to test set labels. SATE follows a two-stage process: in the first step, it calculates subgroup ratios by linearly expressing the average embedding of test data using the average embeddings of each subgroup in a subgroup-labeled training set. In the second step, it estimates subgroup performance using a subgroup-labeled augmented set (or validation set). The final predicted test accuracy is obtained by calculating a weighted sum of subgroup performance from step 2, using the subgroup ratios from step 1. The effectiveness of SATE is demonstrated on both image and language tasks.

**Strengths:**

The paper is clearly written and presents experiments across diverse benchmarks.

**Weaknesses:**

[W1] The rationale for predicting average accuracy based on the test distribution rather than evaluating using worst group accuracy is not clear. Is there a realistic scenario that motivates this? From a group robustness perspective, an ideal model should perform well across all subgroups. For this reason, group robustness studies typically evaluate models using worst-group accuracy or the average performance across subgroups (unbiased accuracy). However, this paper appears to prioritize sample average accuracy, aligned with the test environment distribution, rather than worst-group or unbiased accuracy. The reasoning behind this choice is not well-justified.

[W2] Along with W1, using the labeled set $S'_i$ to measure subgroup performance seems more like conducting a test evaluation than performance prediction. Does assuming access $S'_i$- a labeled set considered unseen from the model’s perspective- appear to be an overly strong assumption?

[W3] For the experiments in Table 1, is the training dataset also composed of corrupted data?

[W4] This method seems to handle only seen subgroups. How does it address unseen subgroups? If the goal is performance prediction, it should ideally be able to handle unseen subgroups as well.

[W5] Obtaining subgroup labels is often costly, and thus many studies have long focused on learning methods that do not require subgroup labels. Requiring a labeled training set for performance prediction appears to set up an unrealistic scenario. This is especially relevant given that even the DFR method used in this paper does not require training set labels during learning.

[W6] How would the approach perform if evaluated using a retrieval-based method? A straightforward solution, for example, could be KNN with $S'_i$.

[W7] Some terms appear in formulas without clear definitions (e.g., $P_{T-emb}$, $P_{g-emb}$, $H_S$)

**Questions:**

Please refer to the weaknesses.

---

> ### Author Response · Authors · 2024-11-25
> **Response to Reviewer qi9w (part 1/2)**
>
> Dear Reviewer qi9w,
>
> We thank you for your valuable feedback and constructive suggestions. Below, we address each of your comments in detail:
>
> > **W1**: The rational for predicting average accuracy is not clear.
>
> **R1**: We address this in the “Subpopulation” paragraph of Section 2 but would like to further emphasize the rationale here:
>
> - Known Test Distribution: Unlike group robustness studies that focus on worst-group accuracy (WGA) to ensure uniform performance across unknown test distributions, our context is performance prediction where the test distribution is known. In such cases, sample average accuracy aligned with the test environment distribution is a more straightforward metric.
> - Broader Implications of Subpopulation Shift: While unfairness (e.g., low worst-group performance) is a critical concern, subpopulation shifts can also cause significant degradation in overall performance. Addressing this issue is equally important, motivating our focus on predicting sample average accuracy.
>
> > **W2**: Does assuming access $S_i'$ appear to be an overly strong assumption?
>
> **R2**:$S_i'$ is the augmented training data from subgroup $i$ if validation data is not available. **This does not need additional information other than training data and an augmenting method.** We do not view this as an assumption, as no existing performance prediction method can operate without access to training data.
>
> > **W3**: Is the training dataset also composed of corrupted data?
>
> **R3**: No, the training dataset is not corrupted. We’ve mentioned “add perturbations to test sets” in the “Real-World Shift” paragraph of Section 5.3. Corrupting both training and test sets is against the goal of evaluating performance prediction methods under covariate shifts.
>
> > **W4**: How does this method address unseen subgroups?
>
> **R4**: Here we develop a lightweight method to **detect unseen subgroups** after the first step of SATE. We use Mean Square Error (MSE) of the linear decomposition as the indicator of the existence of unseen subgroups. Larger MSE indicates higher probability that test set contains unseen subgroup.
>
> We conducted experiments on the NICO++ [1], a commonly used domain generalization benchmark, to evaluate our detection method. The experimental setup and findings are as follows:
>
> - Benchmark Setup: We utilized the NICO++ dataset, focusing on $y \in \{0, 1, 2, 3, 4, 5\}$ and $a \in \{0, 1, 2, 3, 4, 5\}$, resulting in 36 subgroups in total. The training data followed the original split, where subgroup (5,4) was absent. While all 36 subgroups were present in the original test split.
> - Test Sets: To simulate various conditions, we created 50 test sets, each comprising $k$ randomly selected subgroups from the original test set.
> - Evaluation: We evaluate the effectiveness of detection by the Area Under the Curve (AUC) between the existence of unseen subgroup and the MSE of linear decomposition.
>
> |k|5|10|20|
> |-|-|-|-|
> |AUC| 0.950| 0.895| 0.869|
>
> Our results demonstrate that while using linear decomposition to estimate subgroup proportions, MSE is a reliable metric for detecting unseen domains. It consistently performs well when the number of subgroups in the test set becomes large ($k=10, 20$), further extending the applicability of our method to domain generalization scenarios.
>
> [1] https://arxiv.org/abs/2204.08040

---

> ### Author Response · Authors · 2024-11-25
> **Response to Reviewer qi9w (part 2/2)**
>
> > **W5**:  Requiring a labeled training set for performance prediction appears to set up an unrealistic scenario.
>
> **R5**: The requirement for access to training set labels ($y$) is **unavoidable for most performance prediction methods**, such as ATC and NI. Some model-agreement-based methods even need to retrain the model. Similarly, DoC indirectly relies on training set labels as it assumes access to the model’s accuracy on the training set. This reliance on training labels is thus a common requirement across existing approaches.
>
> > **W6**: How would the approach perform if evaluated using a retrieval-based method?
>
> **R6**: Based on our understanding, "KNN with $S_i'$" refers to a approach for estimating test set accuracy. This method identifies the k nearest training subgroups $S_i'$ of the test set $T$ in the embedding space and uses the average accuracy of these k neighbors as the estimated accuracy.
>
> However, this approach may not perform well, as illustrated by the following example: Suppose there are 4 subgroups in total. Test set $T_1$ consists of 20% of subgroup 1 and 80% of subgroup 2, while test set $T_2$ consists of 50% of subgroup 1 and 50% of subgroup 2. For both $T_1$ and $T_2$, the two nearest neighbors will be $S_1'$ and $S_2'$. Consequently, they would produce identical accuracy estimates, despite having different subgroup distributions. This outcome is clearly not reasonable.
>
> If this interpretation does not address your concern, we would appreciate further clarification to ensure an accurate response.
>
> > **W7**: Some terms appear in formulas without clear definitions ($P_\text{T-emb}$, $P_\text{g-emb}$, $H_s$)
>
> **R7**:  $P_\text{T-emb}$ is **already defined in the “Estimating Subgroup Proportion” paragraph of Section 4.2** as the probability distribution of $h_T$, where $h_T$ represents the embedding of a sample from the test set $T$. Similarly, $P_\text{g-emb}$ refers to the embedding distribution for a specific subgroup $g$.
>
> As for $H_s$, it is a temporary variable **defined in Line 1 of Algorithm 1**, it is a $d \times (c \cdot m)$ matrix, where each column corresponds to the average embedding of a specific subgroup. **We have updated the paper and mentioned where $H_s$ is defined when referenced later.**

---

### Official Review · Reviewer_9P9W · 2024-11-04

**Soundness:** 2
**Presentation:** 3
**Contribution:** 2
**Rating:** 6
**Confidence:** 3

**Summary:**

This paper introduces SATE (Subpopulation-Aware Two-stage Estimator), a novel method for predicting model performance under subpopulation shift scenarios, where the distribution of subgroups differs between training and test datasets. SATE's two-stage approach first estimates subgroup proportions in the test set by expressing test embeddings as a linear combination of training subgroup embeddings, then predicts accuracy for each subgroup using augmented training data to produce an overall performance estimate. Experiments show improvement when compared SATE with baselines such as ATC-MC and DoC.

**Strengths:**

1. Novel contribution: First performance prediction method specifically designed for subpopulation shift scenarios and first to address unsupervised performance prediction in NLP tasks.

2. Theoretical foundations: Authors provide proofs of unbiasedness and consistency under certain conditions.

3. Empirical evaluation:  Experiments across multiple domains (vision, medical, NLP) and demonstrates superior performance compared to baselines.

**Weaknesses:**

1. Knowledge of group annotations: the method requires attribute annotations for the training data, which may not always be available or could be costly to obtain.

2. Scalability: The method may struggle with scalability when dealing with a large number of subgroups.

3. Linear decomposition: the method relies on linear decomposition assumption for test set embeddings, which might not always hold.

4. Discussions of limitations: there is no clear discussion of failure modes or performance under noisy/incomplete attribute annotations.

**Questions:**

1. How sensitive is the method to violations of the linear decomposition assumption for test set embeddings?

2. What are the specific conditions required for the theoretical guarantees to hold?

3. What is the memory requirement for storing subgroup embeddings?

4. How robust is the linear equation-solving step when subgroup embeddings are nearly collinear? What happens when some subgroups have very few training samples?

---

> ### Author Response · Authors · 2024-11-25
> **Response to Reviewer 9P9W**
>
> Dear Reviewer 9P9W,
>
> We thank you for your valuable feedback and constructive suggestions. Below, we address each of your comments in detail:
>
> > **W1**: Requires knowledge of group annotations.
>
> **R1**: We acknowledge this limitation. **We have updated the paper and include this limitation in the Limitations Section.** However, in practical scenarios, group annotations can often be feasible in certain contexts, such as when datasets are curated with domain knowledge. Additionally, group labels can be a feature artificially selected from $X$, which users may identify as being responsible for subpopulation shifts. Thus, we believe the need for this additional knowledge is reasonable in subpopulation shift contexts.
>
> > **W2**: Scalability: May struggle when number of subgroups is large.
>
> **R2**: We agree that when the number of subgroups is large and the number of samples per subgroup is small, the law of large numbers (LLN) assumption may break, potentially affecting our method’s performance. However, commonly used subpopulation shift benchmarks do not exhibit a large number of subgroups. Experiments have shown our method’s effectiveness on the CheXpert dataset which has up to 12 subgroups.
>
> > **W3**: Linear decomposition assumption may break.
>
> **R3**: In the paper, we make two key assumptions to ensure the validity of the linear decomposition: Assumption 1 (only subpopulation shift occurs) and Assumption 2 (the embedding matrix is column full rank). In our experiments, **these assumptions hold well, as evidenced by the fact that the test embeddings can be linearly expressed with very high $R^2$ values (>0.99) among all datasets**.
>
> We acknowledge the possibility of unknown or extreme cases where these assumptions may fail. To address this, we have included a discussion of such potential limitations in the revised paper's Limitations section.
>
> > **W4**: No clear discussion of failure modes or performance under noisy/incomplete attribute annotations.
>
> **R4**: We recognize that noisy or incomplete attribute annotations may hurt our method's performance and will mention this limitation in the Limitations Section in the revised paper. Addressing these scenarios is beyond the primary focus of this work.
>
> > **Q1**: How sensitive is the method to violations of the linear decomposition assumption for test set embeddings?
>
> **RQ1**: Please refer to our response in R3.
>
> > **Q2**: What are the specific conditions required for the theoretical guarantees to hold?
>
> **RQ2**: As mentioned in R3, the theoretical guarantees rely on two key conditions: 1.Other kinds of distribution shift between the train and test splits should be mild. 2. The embedding dimensionality should be significantly larger than the number of subgroups to ensure the embedding matrix is column full rank. Commonly used subpopulation shift benchmarks, such as Waterbirds and CelebA, satisfy these conditions, providing practical examples where our method performs effectively.
>
> > **Q3**: What is the memory requirement for storing subgroup embeddings?
>
> **RQ3**: The memory requirement is minimal. Our method only requires **storing the average embedding for each subgroup**. For $k$ subgroups with $d$-dimensional embeddings, the storage cost is $O(kd)$.
>
> > **Q4**: How robust is the linear equation-solving step when subgroup embeddings are nearly collinear? What happens when some subgroups have very few training samples?
>
> **RQ4**: To evaluate the robustness of the linear equation-solving step against collinearity in subgroup embeddings, we compute the Variance Inflation Factor (VIF) for each embedding: $\text{VIF}_i = \frac{1}{1 - R_i^2}$, where $R_i^2$ is the coefficient of determination when regressing embedding $i$ on all other embeddings. A higher VIF indicates stronger collinearity.
>
> The table below summarizes the average VIF values for subgroup embeddings across datasets. Based on these results, we highlight the following observations:
>
> 1. Several datasets in our experiments exhibit moderate collinearity among subgroup embeddings (VIF > 10). Despite this, our linear decomposition approach demonstrates robustness to moderate levels of collinearity.
> 2. None of the datasets show very strong collinearity (VIF > 100), alleviating concerns about perfect collinearity in practical scenarios.
> 3. Higher VIF values are associated with increased errors in estimating subgroup proportions. For instance, the Wasserstein distance between predicted and actual subgroup proportions is higher for the Waterbirds dataset compared to CelebA.
>
> | |Waterbirds_vit|CelebA_vit|MultiNLI_bert|SNLI_bert|
> |-|-|-|-|-|
> |Average VIF|29.9|6.0|22.0|30.5|

---

> > ### Comment · Reviewer_9P9W · 2024-11-28
> >
> > Thank you for the feedback, which answers my questions. I updated my score.

---

> > > ### Author Response · Authors · 2024-12-02
> > >
> > > Thank you very much for reconsidering and increasing your score. Your support has been very helpful to us, and we would be happy to engage in further discussion if needed.

---

### Official Review · Reviewer_uHAx · 2024-11-05

**Soundness:** 3
**Presentation:** 2
**Contribution:** 1
**Rating:** 5
**Confidence:** 4

**Summary:**

The authors tackle the problem of estimating model performance under subpopulation shift. They propose SATE, which estimates test-set group proportions by representing the mean test-set embedding as a convex combination of mean training subgroup embeddings. The test-set accuracy is then a convex combination of the per-group model accuracies. The authors evaluate their method on typical subpopulation shift datasets, finding that they outperform the baselines.

**Strengths:**

- The method is intuitive and easy to understand.
- The authors evaluate their method on the common subpopulation shift benchmarks.

**Weaknesses:**

1. My main concern is regarding the significance of the method. To me, the problem of estimating model performance under subpopulation shift is largely trivial, as it is just a matter of estimating group proportions on the test set. If group labels are provided in the training domain as the authors assume, it is even simpler, and also a much more restrictive problem setup, which limits the applicability of the method. Given that the method is only theoretically bounded when subpopulation shift is the only shift that occurs (Assumption 1), and does not take e.g. the variation of sample difficulty within each subpopulation into account, I am not convinced that this method is useful.

2. It is not surprising that the proposed method outperforms other performance prediction methods (Figure 4), as these baselines are not specific to subpopulation shift, and do not even utilize the training set attributes. There are several other intuitive baselines that the authors could consider, e.g. learning per-group clusters on the training set, learning a debiased group predictor on the training set, or directly learning a model to predict the errors of the original model.

3. The authors should also show the predicted group proportions versus the actual proportions in the appendices.

4. To improve the significance of the work, the authors should consider evaluating their method on domain generalization benchmarks such as DomainBed [1] or WILDS [2].

[1] https://arxiv.org/abs/2007.01434

[2] https://arxiv.org/abs/2012.07421

**Questions:**

1. When computing the test-set group proportion $w$ in Algorithm 1 Step 10, how is it enforced that $w$ should sum to 1?

2. In the result showing augmentations on the y=x line (Figure 2), has the model been trained with the same data augmentations? It seems like this would be an important factor.

---

> ### Author Response · Authors · 2024-11-25
> **Response to Reviewer uHAx (part 1/2)**
>
> Dear Reviewer uHAx,
>
> We thank you for your valuable feedback and constructive suggestions. Below, we address each of your comments in detail:
>
> > **W1.1**: Problem setting is trivial.
>
> **R1.1**: We agree that our method is simple, but we want to highlight our contributions again. Our work is the first to introduce the idea of group proportion estimation to the context of performance prediction. Beyond this, we contribute a novel empirical finding (in Section 4.3), which we leverage to estimate group-wise accuracy effectively. These contributions collectively provide a **new perspective** on performance prediction.
>
> > **W1.2**: Problem setup is more restrictive because we need group annotations.
>
> **R1.2**: We acknowledge this limitation. **We have updated the paper and mention this limitation in the Limitations Section.** However, in practical scenarios, group annotations can often be feasible in certain contexts, such as when datasets are curated with domain knowledge. Additionally, group labels can be a feature from $X$, which users may identify as being responsible for subpopulation shifts.
>
> > **W1.3**: Assume subpopulation shift is the only shift that occurs.
>
> **R1.3**: Our method is primarily designed for subpopulation shifts, but it is also **robust to moderate covariate shifts in practice**, as demonstrated in Table 1 in our paper. And we have already discussed this limitation in the Limitations Section.
>
> > **W2.1**: It is not surprising that the proposed method outperforms other performance prediction methods since they do not utilize the training attributes.
>
> **R2.1**: We agree with you that our method utilizes more information, but not using attributes is not an excuse for current baselines to perform poorly in subpopulation shift scenarios, as shown in our experiments. **Our work highlights this gap in the field and proposes a simple yet effective approach to address it.**
>
> > **W2.2**: Intuitive Baselines should be compared.
>
> **R2.2**: We agree that some of these intuitive ideas are reasonable.
> 1. **Learning a debiased group predictor.** This can only serve as an
> **alternative for the first step** of our approach (proportion estimation) rather than a baseline of SATE.
> We compared our linear decomposition (LD) method with the debiased group predictor (GP) using both Wasserstein distance and cross-entropy between the ground truth and estimated subgroup distribution.
>
>     |Wasserstein Distance$(\downarrow)$| Waterbirds | CelebA | CheXpert | MultiNLI | SNLI|
>     | - | - | - | - | -| -|
>     | LD (ours) | 0.053 $\pm$ 0.039| **0.039** $\pm$ 0.031| **0.028** $\pm$ 0.008| **0.049** $\pm$ 0.019| **0.065** $\pm$ 0.023|
>     | GP | **0.050** $\pm$ 0.029| 0.043 $\pm$ 0.026| 0.050 $\pm$ 0.07| 0.050 $\pm$ 0.019| 0.093 $\pm$ 0.032|
>
>     |Cross Entropy$(\downarrow)$| Waterbirds | CelebA | CheXpert | MultiNLI | SNLI|
>     | - | - | - | - | -| -|
>     | LD (ours) | 1.22 $\pm$ 0.15| **1.19** $\pm$ 0.16| 2.47 $\pm$ 0.02| **1.66** $\pm$ 0.25| **2.26** $\pm$ 0.36|
>     | GP | **1.20** $\pm$ 0.18| 1.22 $\pm$ 0.22| 2.47 $\pm$ 0.02| 1.89 $\pm$ 0.24| 3.46 $\pm$ 0.76|
>
>     Our results show that LD slightly outperforms GP, while its time complexity is significantly smaller than that of GP. If we have $n$ training samples, $k$ subgroups and $d$ dimensional embeddings, time complexity of LD is $O(k^2d)$ and time complexity of GP is $O(nd^2)$. For Resnet architecture and CelebA dataset, $n=19000,k=4,d=2048$.
>
> 1. **Directly learning a model to predict the errors of the original model.** Since our problem setting focuses on unsupervised accuracy estimation, we have no access to (dataset, error) pairs other than the training set and training error, so it is infeasible to directly train a model to predict the error of the original model. A possible method to get these pairs may be to split the original training set, retrain several models and get their errors on the reserved part. But this retraining approach requires access to the training details and architecture of the original model, making it less applicable to real world settings.
>
> > **W3**: The authors should also show the predicted group proportions versus the actual proportions.
>
> **R3**: We have revised the paper and include the Wasserstein distance and cross entropy between the predicted and actual subgroup proportions in the appendix, which provides a quantitative measure of the dissimilarity between them. For clarity, here we randomly select three pairs of predicted and actual proportions from the Waterbirds dataset to illustrate the comparison.
>
> | predicted proportions| actual proportions|
> |-|-|
> |0.27,0.27,0.23,0.23|0.25,0.25,0.25,0.25|
> |0.32,0.31,0.18,0.19|0.30,0.30,0.20,0.20|
> |0.17,0.19,0.32,0.32|0.15,0.15,0.35,0.35|

---

> ### Author Response · Authors · 2024-11-25
> **Response to Reviewer uHAx (part 2/2)**
>
> > **W4**: Should evaluate on domain generalization benchmarks.
>
> **R4**: Here we develop a lightweight method to **detect unseen subgroups** after the first step of SATE. We use Mean Square Error (MSE) of the linear decomposition as the indicator of the existence of unseen subgroups. Larger MSE indicates higher probability that test set contains unseen subgroup.
>
> We conducted experiments on the NICO++ [1], a commonly used domain generalization benchmark, to evaluate our detection method. The experimental setup and findings are as follows:
>
> - Benchmark Setup: We utilized the NICO++ dataset, focusing on $y \in \{0, 1, 2, 3, 4, 5\}$ and $a \in \{0, 1, 2, 3, 4, 5\}$, resulting in 36 subgroups in total. The training data followed the original split, where subgroup (5,4) was absent. While all 36 subgroups were present in the original test split.
> - Test Sets: To simulate various conditions, we created 50 test sets, each comprising $k$ randomly selected subgroups from the original test set.
> - Evaluation: We evaluate the effectiveness of detection by the Area Under the Curve (AUC) between the existence of unseen subgroup and the MSE of linear decomposition.
>
> |k|5|10|20|
> |-|-|-|-|
> |AUC| 0.950| 0.895| 0.869|
>
> Our results demonstrate that while using linear decomposition to estimate subgroup proportions, MSE is a reliable metric for detecting unseen domains. It consistently performs well when the number of subgroups in the test set becomes large ($k=10, 20$), further extending the applicability of our method to domain generalization scenarios.
>
> > **Q1**: How is it enforced that $w$ should sum to 1?
>
> **RQ1**: Based on our assumptions, the weights $w$ theoretically sum to 1 without external enforcement. In practice, we normalize $w$ after solving the linear equation.
>
> > **Q2**: In figure 2, has the model been trained with the same data augmentations?
>
> **RQ2**: The models were trained without data augmentation. This decision aligns with the standard practices followed in SubpopBench [2], where no data augmentation is applied during training.
>
> [1] https://arxiv.org/abs/2204.08040
>
> [2] https://arxiv.org/abs/2302.12254

---

> > ### Comment · Reviewer_uHAx · 2024-11-26
> >
> > Thank you for the response. The new experiments have addressed some of my concerns, and I have raised my score to a 5 as a result. However, I believe that W1.1-W1.3 are still fundamental weaknesses of the paper that limit its significance.

---

> > > ### Author Response · Authors · 2024-12-02
> > >
> > > Thank you very much for reconsidering and increasing your score. We really appreciate the discussion and your valuable feedback. We would be happy to engage in further discussion if needed.

---

### Meta-Review · Area_Chair_myBC · 2024-12-21

**Metareview:**

This paper introduces SATE (Subpopulation-Aware Two-stage Estimator), a novel method for predicting model performance under subpopulation shift scenarios, where the distribution of subgroups differs between the training and test datasets. SATE employs a two-stage approach: first, it estimates the subgroup proportions in the test set by representing test embeddings as a linear combination of training subgroup embeddings; second, it predicts the accuracy for each subgroup using augmented training data to produce an overall performance estimate.

However, several concerns have been raised regarding the significance and practicality of the method:

Triviality of the problem: Estimating model performance under subpopulation shift is argued to be a relatively straightforward task, particularly if subgroup labels are available in the training domain, as assumed by the authors. This setup simplifies the problem significantly and makes it much more restrictive, thereby limiting the method's applicability.

Limited theoretical scope: The method is theoretically grounded only under the assumption that subpopulation shift is the sole type of distribution shift (Assumption 1). It does not account for other factors, such as variations in sample difficulty within subpopulations, raising doubts about its utility in more complex real-world scenarios.

Baseline comparisons: While SATE outperforms other performance prediction methods (Figure 4), this result is not unexpected, as the baselines are not tailored for subpopulation shift and do not leverage training set attributes. The paper misses the opportunity to compare against more intuitive baselines, such as learning per-group clusters on the training set, training a debiased group predictor, or directly modeling the prediction errors of the original model.

These limitations suggest that while SATE offers an interesting approach, its broader significance and utility remain unconvincing, particularly in scenarios beyond the constrained setup considered in the paper.

**Additional Comments On Reviewer Discussion:**

The authors acknowledge that their method is relatively simple, despite being the first to introduce the concept of group proportion estimation within the context of performance prediction. The improved performance prediction achieved by leveraging group-wise accuracy estimation is fairly straightforward. Additionally, the authors recognize that the problem setup is more restrictive, as it requires group annotations, which limits the method's broader applicability.

Following the rebuttal, the reviewers did not express significant support for the paper.

---

### Decision · Program_Chairs · 2025-01-22

Reject